# Atomistic Investigation on the Blocking Phenomenon of Crack Propagation in Cu Substrate Reinforced by CNT

**DOI:** 10.3390/nano13030575

**Published:** 2023-01-31

**Authors:** Jee Soo Shim, Hyeon Gyu Beom

**Affiliations:** Department of Mechanical Engineering, Inha University, 100 Inha-ro, Incheon 22212, Republic of Korea

**Keywords:** molecular statics simulation, carbon nanotube, copper, composites, crack propagation, fracture mechanism, fracture toughness

## Abstract

Recently, many researchers in the semiconductor industry have attempted to fabricate copper with carbon nanotubes for developing efficient semiconductor systems. In this work, tensile tests of a carbon-nanotube-reinforced copper specimen were conducted using the molecular statics method. The copper substrate utilized in the tensile tests had an edge half-crack, with the carbon nanotube located on the opposite side of the copper substrate. Subsequently, the effects of carbon nanotube radius were investigated. The mechanical properties of the copper/carbon nanotube composite were measured based on the simulation results, which indicated that the atomic behavior of the composite system exhibited the blocking phenomenon of crack propagation under tension. The fracture toughness of the composite system was measured using the Griffith criterion and two-specimen method, while the crack growth resistance curve of the system was obtained by varying the crack length. This study demonstrated that the mechanical reliability of copper can be improved by fabricating it with carbon nanotubes.

## 1. Introduction

The semiconductor industry is facing new turbulence owing to the limitations of the existing two-dimensional design method. Recently, the semiconductor industry has been attempting to design semiconductors in three dimensions (3Ds); as a result, the importance of 3D electronic packaging has also been highlighted [1]. Stacking semiconductors in 3Ds through silicon via (TSV) technology (not previously used) is being used currently [2]. When TSV technology is used, copper (Cu) is generally filled in a space through which silicon can penetrate for electric conduction. Cu has been widely used for electrical conduction in the past and has good electrical properties and low cost. However, Cu also has many disadvantages, and researchers are attempting to compensate for them. Generally, researchers have focused on carbon nanotubes (CNTs) with excellent mechanical and electrical properties [3,4], with many attempts made to fabricate them with Cu. Consequently, many researchers have discovered the possibility of using Cu/CNT composites. Subramaniam et al. [5] reported that Cu/CNT composite materials have an ampacity 100 times higher than that of pure Cu, which is a conventional conductor. Feng et al. [6] developed a process to utilize Cu/CNT composite materials as the interconnect materials of TSV.

However, these studies focused only on improving the performance of the conducting material, Cu. These experimental results did not consider the degradation of the mechanical reliability caused by cracks that may occur in 3D electronic packaging materials. In general, in electronic packaging materials, cracks may occur owing to continuous driving, and degradation of the product’s properties can be observed accordingly. Therefore, a fracture mechanics perspective is required for improving the reliability of a product. Additionally, many researchers have difficulty with the existing knowledge alone because of the nonlinearity of nanomaterials. Thus, a new analysis method is required because nanomaterials exhibit nonlinear and discontinuous properties not observed in existing bulk materials or microunit materials [7].

In addition, conventional experimental methods have many limitations in terms of finely observing the movement of the atomic units of nanomaterials. In this case, a new observation method is required, and an atomistic simulation may be suitable. Utilizing atomic computer simulations, the atomic behaviors at the crack tip of the Cu/CNT composite material can be observed easily and in detail. Many researchers have attempted to analyze the nonlinear behavior of nanomaterials using atomistic simulations. Recently, it was confirmed that crack penetration, contrary to the existing theory, occurred in an aluminum/silicon vertically cracked nanofilm/substrate system with a coherent interface when the system was under tension [8]. Silicon generally exhibits brittle properties, but it has been confirmed that silicon nanowires of a certain size or less cause ductile transition [9]. Additionally, it was also confirmed that the ductile transition section expanded further when the silicon nanowire was wrapped by CNT [10]. Based on these results, an atomistic simulation is a useful tool for comprehensively analyzing the nonlinear behavior of nanomaterials.

Until now, research on Cu/CNT composite materials has mainly focused on the performance and efficiency of devices through experimental methods, while research on mechanical reliability from a fracture mechanics perspective is insufficient. Therefore, in this study, a fracture mechanics analysis was conducted on a Cu/CNT composite specimen with a half-crack through molecular statics (MS) simulation [8]. The effect of CNT size in the composite material on the mechanical reliability of the entire composite material was analyzed, and the fracture toughness of the Cu/CNT composite material was evaluated. In addition, crack propagation in the Cu/CNT composite was observed in atomic units, and the effect of atomic behavior on the mechanical reliability of the system was confirmed. Overall, this study verified the mechanical reliability of Cu/CNT composite materials and the reliability improvement mechanism through atomistic simulations.

## 2. Details of Molecular Statics Simulations

### 2.1. Empirical Potentials for Describing Atomic Interactions

In this study, the atomic computer simulation program LAMMPS (version of 16Mar2018, Sandia National Laboratories) was used to perform an atomic-scale tensile test of a Cu/CNT composite material [11]. In general, the reliability of the numerical analysis simulation data was verified through a comparison with experimental and theoretical methods. For the reliability of atomic computer simulation data, many researchers have obtained empirical potential energies that set the interactive forces between atoms through experiments and quantum mechanics-based ab initio methods [12]. In this study, three types of interatomic force were established: Cu-Cu, Cu-C, and C-C. The embedded atom method (EAM) potential developed by Mishin et al. [13] was used to reproduce the metallic bonds between the Cu atoms. 

The second-generation reactive empirical bond-order (REBO) potential was adopted to reproduce the bond between the C atoms of the honeycomb structure [14,15]. The equation of the REBO potential is as follows:(1)$$V^{\mathrm{REBO}}=\sum_{i}\sum_{j<i}\left[V_{R}(r_{ij})+\kappa_{ij}V_{A}(r_{ij})\right],$$
where:(2)$$V_{R}(r_{ij})=f_{ij}(1+Q_{ij}/r_{ij})H_{ij}e^{-\alpha_{ij}r_{ij}},$$
(3)$$V_{A}(r_{ij})=-f_{ij}\sum_{n=1,3}b_{ij}^{\left(n\right)}e^{-\beta_{ij}^{\left(n\right)}r_{ij}},$$
where $V_{R}$ and $V_{A}$ indicate repulsive and attractive functions, respectively. The parameter $r_{ij}$ indicates the distance between *i* and *j* atoms. Other potential parameters are described in detail in the reference paper [14]. The original REBO potential has a switching term in the cutoff function $f_{ij}$, which shows the abnormal hardening of CNT under tension [16]. In this study, a simulation was conducted while removing the switching term. The cutoff distance in the REBO potential was set to 0.2 nm in this study. The stiffness of Cu and Young’s modulus of the CNT calculated utilizing the EAM and REBO potentials are listed in Table 1. In the table, the calculation results of the mechanical properties and cohesive energy using the empirical potential showed reasonable results when compared with experimental values.

The Lennard–Jones (LJ) potential was used to reproduce the van der Waals force between a Cu atom and C atom. The LJ potential equation is as follows [21]:(4)$$V_{ij}^{LJ}=4\varsigma \left[{\left(\psi /r_{ij}\right)}^{12}-{\left(\psi /r_{ij}\right)}^{6}\right],$$
where the potential depth ς and equilibrium separation ψ were set to 0.02578 eV and 0.30825 nm, respectively [22,23]. In addition, the cutoff parameter for Cu-C atomic interactions was set to 0.770625 nm [24].

### 2.2. Details of the Simulation Models

The atomic configuration of a representative simulation model is shown in Figure 1. Based on Cartesian coordinates, the thickness *t* and width *w* directions of the Cu specimen are *x* and *y* directions, respectively. The axis direction of the CNT in the simulation model is in the *z*-direction, equivalent to the height *h* direction of the Cu specimen. Cu has a face-centered cubic (FCC) structure. The crystallographic orientation of the Cu specimen was set to the most basic [100], [010], and [001] directions based on the Cartesian coordinate system. The lattice parameter a of Cu was set to 0.361 nm. Based on Cartesian coordinates, each length of the simulation model is described as w = 90a, h = 90a, and t = 15a, respectively. At half the height of the specimen, the (001)/[010] (plane/front) half-crack in the Cu specimen was created by removing the interaction between the Cu atoms. The hole of the Cu substrate containing the CNT was made by removing atoms in a cylindrical shape. The center of the Cu hole axis and the center of the CNT axis were set to be the same and merged. The center of the CNT axis in the Cu specimen is located at *y* = 0.75*w* = 67.5*a* and *x* = 0.5*t* = 7.5*a* of the simulation model based on origin *O*. The direction of the crack and axial direction of the CNT were perpendicular to each other. Table 2 shows the coordinates of the CNT used in this study and their corresponding radii. Only armchair CNTs, whose chiral vector is expressed as (*n*,*n*), were used in this study to simplify the results. Additionally, a pure Cu specimen without CNT was used in this study to compare the results with those of the Cu/CNT composite model.

A periodic boundary condition was applied in the thickness *t* direction of the simulation model, and a non-periodic boundary condition was applied in the other directions. Thus, the simulation was conducted under plane strain conditions. As a result, CNT exists indefinitely at regular intervals in the *t* direction. A similar model has been used in another study [25]. The tensile tests of the simulation model were conducted using the MS method [8]. The conjugate gradient method is the optimization technique used in the MS method [26]. The tolerance parameter for optimization was set to 10^−24^ eV/nm. For the tensile test, both ends of the Cu substrate in the *h* direction were set as boundary regions using a 1 Cu atomic lattice parameter. Each boundary region was pulled by a strain set in opposite directions. To prevent problems caused by the strain being too fast, this study adopted the strain applied to the simulation models as 0.0005. Unlike the molecular dynamics method, the MS method has the advantage of preventing problems caused by high strain rates. The simulation temperature was set to 0 K. All simulation processes were observed using the OVITO program [27].

## 3. Results

### 3.1. Stress–Strain Curves

The stress curves corresponding to the tensile strain are plotted, as shown in Figure 2, to analyze the tensile load reaction of the atoms constituting the Cu/CNT composite system. The average stress tensor in the atomic computer simulations was calculated using the virial stress equation [28]. A previous study [29] confirmed that the virial stress in atomic simulation models can be treated as the Cauchy stress of a continuous solid. In the figure, the tensile stress direction is along the *h* direction, while the tensile strain direction was equivalent to that of tensile stress. Among the various types of strain, the tensile strain in this figure is the engineering strain. In the figure legend, the (*n*,*n*) coordinate indicates the chiral vector of the CNT in the composite.

Overall, the pure Cu specimen without CNT under tension exhibited increasing and falling stages. In the increasing stage, the inclination shape of the stress curve was linear. When the strain of the pure Cu specimen reached a critical value, the corresponding stress curve exhibited a drastic decrease. In this study, the failure strain of the pure-Cu specimen was 0.042. Failure strain was measured using the strain value at the maximum stress. The drastic decrease in the stress curve indicated that the crack in the pure Cu specimen propagated unstably under uniform tension, generally reported as a characteristic of a Cu specimen with a nanocrack under uniform tension based on a previous study [30]. Thus, nanocracks in the system can affect the mechanical reliability of the entire system.

However, when CNTs were added to the Cu specimen, the stress curve showed a significant change. The stress curve of the Cu/CNT composite could be classified into three stages: increasing, falling, and undulating. Unlike the clean and linear stress curves of the pure Cu specimen, the stress curves of the Cu/CNT composite showed an overall vibrating behavior. In the increasing stage, the undulating and linear stress curve is caused by the hyperelastic characteristics of the CNT [31]. As CNTs with hyperelastic properties were captured by the Cu specimen, the stress curve exhibited an undulating and linear form without showing hyperelastic properties. After the strain of the Cu/CNT system reached a critical value, a drastic decrease began. The failure strain of all Cu/CNT composite models was 0.047 but was improved by approximately 111% after adding CNTs to a Cu specimen in this study. In the drastic falling stage of the stress, the cracks in the composite grew unstable. However, the drastic decrease in the stress curve soon stopped and began an undulating process. In the undulating process, unstable crack propagation in the system was no longer observed, but initiated ductile deformation of the Cu, which is related to the deformation behavior of the Cu/CNT composite (discussed in the Section 3.2). The stress curves of the Cu/CNT composite and pure Cu specimen, especially the undulating process of the composite, indicate that the energy required to break the Cu/CNT composite was improved compared to that of the pure Cu specimen. Consequently, the mechanical reliability of Cu can be improved by being fabricated with CNT, while preventing a sudden fracture of the system caused by a nanocrack. In Figure 2, when the radius of the CNT in the system increases, the undulating position of the stress curve also increases, based on the *y*-axis; this phenomenon is discussed in the Section 3.2.

Young’s modulus of the composite was investigated by comparing the simulation results with additional calculations to verify the mechanical reliability of the Cu/CNT system from the simulation results. Young’s modulus of the system was measured by regression analysis of the linearly increasing stage in the stress–strain curve. Young’s modulus *Y*_MS_ of the Cu/CNT composite measured in this study is summarized in Table 3. In this study, Young’s modulus of a pure Cu specimen with a nanocrack was measured to be approximately 50.688 GPa. The experimental Young’s modulus of Cu, whose crystallographic orientation is aligned along the [001] direction, was reported to be approximately 59–77 GPa [32]. This difference between the simulation and experimental values was caused by a nanocrack in the Cu specimen. As mentioned previously, nanocracks in the system cause major degradation in the mechanical reliability of the entire system. In addition, it was confirmed in a previous study that the yield strength and Young’s modulus of Cu decreased as the length of the crack increased [30]. However, Young’s modulus of the Cu/CNT composite was improved compared with that of the pure Cu specimen. The Young’s modulus of the Cu/CNT composite, including the (4,4) CNT, was measured to be approximately 52.877 GPa. As the radius of the CNT in the composite increased, Young’s modulus of the composite increased. This phenomenon is caused by the increasing area of CNT with excellent mechanical properties in the composite, as shown in Table 3. However, as the radius of the CNT in the composite increased, Young’s modulus of the composite tended to converge to some extent. The Young’s modulus of the Cu/CNT composite, including the (19,19) CNT, was measured to be approximately 59.859 GPa. In this study, when the CNTs were added to a Cu specimen with a nanocrack, Young’s modulus of the system showed an improvement rate of approximately 104–118%.

An additional Young’s modulus calculation was conducted to verify the simulation results. Young’s modulus of the composite system can be expressed by the rule of mixtures [33]:(5)$$Y_{RM}=(Y_{1}A_{1}+Y_{2}A_{2})/(A_{1}+A_{2}),$$
where *Y* and *A* denote Young’s modulus and *x*-*y* plane area of the material in this study, respectively. The subscript number denotes the material constituting the composite system. The parameters for calculating Young’s modulus of the Cu/CNT composite according to Equation (5) are summarized in Table 3. The thickness of the CNT, an important element of the *x*-*y* plane area of the CNT, was set as 0.34 nm [34]. Considering the CNT thickness *s*, the gap between the hole and CNT in the system was set to exceed 0.17 nm. The cross-section of the CNT was calculated using the formula 2π*rs*, while the cross-section of the Cu specimen was calculated by subtracting the area of an internal hole from the *x*-*y* plane section: *wt* − π*R*^2^. The *Y*_Cu_ was measured by regression analysis of the stress curve which is obtained by tensile testing a model with only hole and crack without CNT. In the table, it was confirmed that Young’s modulus *Y*_RM_, which is calculated by Equation (5), is consistent with the MS simulation value *Y*_MS_. Thus, the cross-section of the specimen made of atoms was assumed to be a continuum with an appropriate size: 2π*rs* + *wt* − π*R*^2^.

### 3.2. Deformation Behaviors of the Cu/CNT Composites

In this study, the atomic configurations of the simulation models were plotted to analyze the deformation behaviors of the Cu/CNT composites under tension, particularly after crack propagation. In these configurations, the color of the atoms in the simulation model depicts the potential energies of each atom. The numbers next to the color bar of the configuration indicate their ranges. The atomic configuration depicting the fracture behavior of the pure Cu specimen is plotted in Figure 3. The potential energy of the atom consisting of the pure Cu specimen ranged from −3.54 eV to −2.52 eV. Before the specimen reached the critical strain, the Cu atoms moved stably in the tensile direction. As the tensile strain of the pure Cu specimen reached 0.042, the crack in the specimen propagated through the interatomic bonds of one layer at the crack tip, indicated by the red box in the figure. Subsequently, the crack propagation of the specimen showed unstable manner because of debonding between atoms at the crack tip. When the tensile strain reached 0.0445, unstable crack propagation ended, and the Cu specimen was completely broken. The newly formed plane of the fractured Cu specimen was clean. The crystallographic orientation of the newly formed plane corresponded to the (001) plane. The Cu specimen with a nanocrack under tension usually showed a clean plane caused by unstable crack growth, as reported in a previous study [35]. 

However, the deformation behaviors of the system differed when CNTs were added to the Cu specimen. The atomic configuration depicting the deformation behavior of the Cu/CNT composite is shown in Figure 4. In the Cu/CNT composite, the chiral index of the CNT was (4,4). In the deformation process of the (4,4) model, the potential energy of the atom ranged from −7.6 eV to −2.52 eV. Similar to the deformation behaviors of the pure Cu specimen, the crack in the Cu/CNT composite began propagating when the strain reached a critical value. As the tensile strain of the Cu/CNT composite reached 0.047, the interatomic bonds of one layer at the crack tip broke. The crack in the system initiated propagation owing to the broken crack in one layer of atoms. However, when the crack reached the vicinity of the CNT, crack propagation behavior was delayed. As the strain of the Cu/CNT composite containing (4,4) CNT reached 0.048, the unstable manner of crack growth stopped. Owing to the blocking of the CNT, the crack cannot propagate unstably and initiates detouring near the CNT. The reason that cracks in the Cu cannot penetrate the CNT is related to the higher stiffness of the CNT compared to that of Cu [36]. As the strain of the Cu/CNT composite reached 0.049, the detouring of the crack in the specimen was complete. As the tensile strain of the model reached 0.076, the crack tip was V-shaped and blunted. When the strain reached 0.0765, a new crack was formed in the blunted crack. After new crack formation in the blunted crack, the Cu/CNT composite showed continuous ductile deformation. The Cu specimen slips continuously in two directions—the (011) plane direction and the (01$\bar{1}$) plane direction—resulting in ductile deformation. Moreover, the ductile deformation of the Cu specimen exhibited a combination of slip and twinning plastic deformation [37]. 

As shown in Figure 2, the undulating stage in the stress curve shows a higher position when the radius of the CNT in the model increases. The atomic configurations depicting the deformation behaviors of the Cu/CNT composite containing (19,19) CNT are plotted in Figure 5. Similar to the (4,4) model, the potential energy of the atom ranged from −7.6 eV to −2.52 eV in the deformation process of the (19,19) model. Similar to the deformation behaviors of the (4,4) model, the interatomic bonds at the crack tip broke as the strain on the composite reached a critical value. In this case, unstable crack growth in the specimen ended when the tensile strain of the composite reached 0.0485. Unlike the deformation behavior of the (4,4) model, the path for the crack to propagate becomes narrower in this case, resulting in the Cu/CNT composite containing the (19,19) CNT immediately showing ductile deformation, not detouring the CNT; ductile deformation of Cu in the (19,19) model continued thereafter. As the tensile strain of the (19,19) model reached 0.15, the V-shaped crack detoured the CNT in half. Observing the deformation behaviors of the Cu/CNT composites, it can be confirmed that the rapid fracture caused by a nanocrack in the Cu specimen can be prevented by adding CNT to the Cu specimen.

### 3.3. Failure Properties of the Cu/CNT Composite System

Observing the stress curves and deformation behavior of the Cu/CNT composite, it was confirmed that the mechanical reliability of the Cu specimen could be improved by adding CNT to the Cu substrate. To analyze the reliability improvement of the Cu/CNT composite, its failure properties were investigated. The failure strength of the Cu/CNT system, which is a representative failure property, was also investigated. The failure strength was set as the maximum stress in the stress–strain curve, which is the onset of crack propagation. The measured values corresponding to the various radii of the CNT are presented in Table 4. 

The table confirms that the failure strength of the Cu/CNT composite was higher than that of the pure Cu specimen. The failure strength of the pure Cu specimen was measured as approximately 2.098 GPa. Additionally, it was confirmed that the failure strength of the system improved when CNTs were added to the Cu specimen. The failure strength of the Cu/CNT composite containing (4,4) CNT was measured to be approximately 2.471 GPa. The failure stress of the simulation model shows some agreement with the experimental values [38]. The reason why the result values show a slight difference is because the Cu/CNT composite model in the simulation is not completely the same as the composite material structure in the experiment. In Cu in the experiment, relatively low mechanical properties may be measured due to polycrystalline structures and more defects in the specimen. Moreover, it was confirmed that as the radius of the CNT increased, the failure strength of the composite also increased because the volume ratio of CNT with excellent mechanical properties in the Cu/CNT composite increased. However, it was confirmed that the failure strength of the composite did not increase continuously and converged to approximately 2.7 GPa as the radius of the CNT increased, related to the fact that the volume ratio of CNT that can be increased within the simulation model is limited to some extent. In particular, the amount by which the radius of the CNT can increase is limited by the length in the *t* direction of the simulation model. Through this, it was confirmed that the failure strength of the Cu specimen can be improved by adding CNT.

Analyzing the failure strength of the Cu/CNT composite, it was found that the mechanical reliability of Cu could be improved by adding CNT. However, the design using the failure strength described above is not satisfactory from a conservative perspective because the CNTs are not broken. The fracture toughness of the Cu/CNT system was measured in this study to obtain a more conservative design. The fracture toughness of the system was measured using the Griffith criterion [39] and the two-specimen method [40]. Based on the Griffith criterion, the critical energy release rate $G_{c}$ can be described by the surface energy $\gamma_{s}$, and its equation is as follows:(6)$$G_{c}=2\gamma_{s}.$$

The surface energies for each simulation model are listed in Table 4. The surface energy of the (001) plane in the pure Cu specimen, based on the Griffith criterion, was measured to be approximately 1.342 J/m^2^. The simulation results in this study tended to be somewhat consistent with that of other study [30]. It was confirmed that the surface energy of the system tended to increase when CNTs were added to the Cu specimen. The cross-sectional area of the specimen used to calculate the surface energy was set to be the same as that used to calculate Young’s modulus of the system: 2π*rs* + *wt* − π*R*^2^. When (4,4) CNTs were added to the Cu specimen, the surface energy of the system was measured to be approximately 1.355 J/m^2^. The surface energies of the system containing (7,7) and (10,10) CNT were measured to be 1.368 J/m^2^ and 1.377 J/m^2^, respectively. The surface energies of the other models were measured as approximately 1.394 J/m^2^, 1.402 J/m^2^, and 1.412 J/m^2^, respectively. As the radius of the CNT increases, the hole size inside the system increases; however, the surface energy of the composite increases because of the excellent shell edge energy of the CNT. The experimental value with respect to the shell edge energy of the CNT was reported to be approximately 8 J/m^2^ [41]. Thus, fracture toughness could possibly be increased by adding CNTs to the Cu specimen.

An additional fracture-toughness calculation of the system was conducted to verify the simulation data. As previously mentioned, the two-specimen method was used to calculate the critical energy release rate of the system [40]. The equation for the two-specimen method is as follows:(7)$$G_{c}^{t}=-\frac{U(c+\Delta c)-U(c)}{\Delta c\cdot t},$$
where *U* denotes the strain energy, and *c* and *t* denote the crack length and thickness of the specimen, respectively. The two-specimen method was used to depict the crack growth resistance curves of the Cu/CNT composite by varying the crack length. The additional crack length Δc used to calculate the energy release rate of the material was set as 1*a,* which is the lattice parameter of Cu. A pre-crack was set near the CNT to confirm the variation in the energy release rate around the CNT in the specimen. The lengths of the pre-crack were set as *c* = 23.465 nm to *c* = 25.27 nm based on the width *w* direction, which is the same as the 2.5*a* apart in both directions, based on the center of the CNT axis. Simulation models containing (4,4) and (7,7) CNT were utilized to obtain the crack growth resistance curves. 

Before discussing the crack growth resistance curves, the strain energy of the system in the elastic section corresponding to the tensile strain is discussed to understand the phenomenon. The strain energy curves of each set of crack length models are plotted in Figure 6 and Figure 7. From the figures, it was confirmed that the inclination of the strain energy curves decreased when the crack length in the system increased. In addition, it was confirmed that both strain energy curves have a small gap between the *c* = 24.548 nm and *c* = 24.909 nm models, attributed to the crack-tip position. In the *c* = 24.548 nm model, the crack tip of the system was still located on the inner side of the CNT. As CNTs with excellent properties can still withstand tensile loads, the *c* = 24.548 nm model has a higher strain energy value than the *c* = 24.909 nm model. Otherwise, the crack tip of the *c* = 24.909 nm model was located on the Cu specimen based on the *w* direction. In the *c* = 24.909 nm model, Cu remains the only element that can withstand the tensile load throughout the system, so it shows a lower strain energy curve.

Therefore, the crack growth resistance curve of the Cu/CNT composite showed an inflection point. The critical energy release rate values of each simulation model, calculated using the two-specimen method, are plotted in Figure 8. The critical energy release rate based on the Griffith criterion is also plotted in the figure for comparison. The *x*-axis of the graph, which was set as the crack length, was based on the varying *c* value. In the figure, it can be observed that the energy release rate of the crack length *c* < 24.548 nm model and the energy release rate according to the Griffith criterion showed similar values. Based on this result, the cross-sectional area used to calculate the Griffith criterion can be assumed to be an appropriate value. However, because the energy required to penetrate the CNT is consumed, an inflection point is observed in the trend of the critical energy release rate. The critical energy release rate values of the *c* = 24.548 nm model containing (4,4) CNT and (7,7) CNT were measured to be approximately 10.076 J/m^2^ and 15.123 J/m^2^, respectively. After the inflection point, the energy release rates of the simulation models showed values similar to those of the Griffith criterion. The critical energy release rate values of the *c* = 24.909 nm model containing the (4,4) CNT and (7,7) CNT were measured to be approximately 2.845 J/m^2^ and 2.8 J/m^2^, respectively. Thus, the Griffith criterion could be used to design the reliability of the Cu/CNT composite material. However, for a more accurate design, measurements using the two-specimen method near the CNT are also required.

## 4. Conclusions

In this study, tensile tests of a Cu specimen reinforced by CNT were conducted. The Cu specimen had an edge half crack, with the CNT axis located at a quarter of the width of the simulation model opposite the edge crack. The mechanical reliability of Cu/CNT composites was also investigated. Without CNT, the pure Cu specimen under tension showed a clean plane. However, the cracks in the Cu/CNT composite exhibited delayed propagation. Thus, CNT can improve the mechanical reliability of Cu by being added to Cu. In addition, the fracture toughness of the Cu/CNT composite was investigated to obtain a more conservative design. In addition, the Griffith criterion and two-specimen method were utilized, and the calculated values are discussed. The results indicated that it was necessary to use a more accurate cross-sectional area to calculate the fracture toughness of the system via the Griffith criterion, while the two-specimen method was required near the CNT for more accurate critical energy release rate measurements.

## Figures and Tables

**Figure 1 nanomaterials-13-00575-f001:**
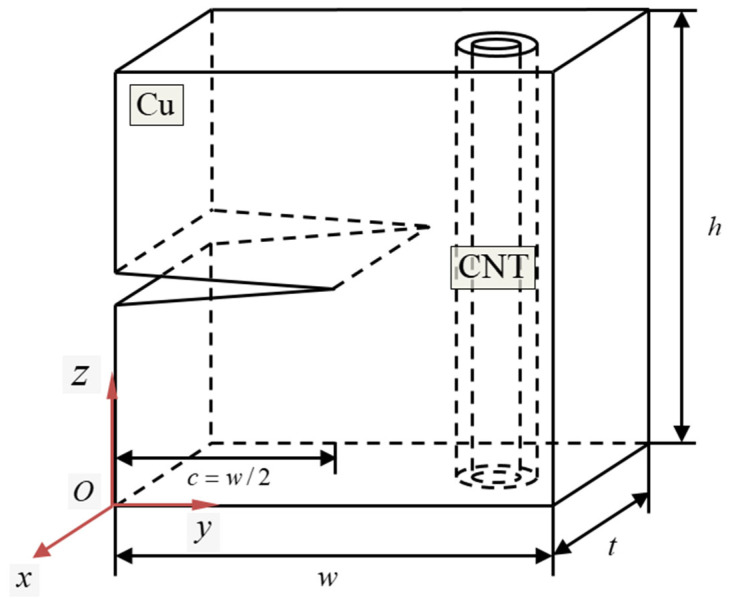
Geometric schematic of the simulation model. Based on the Cartesian coordinates, the crystallographic orientations of the Cu substrate are aligned along the [100], [010], and [001] directions, respectively. The (001)/[010] (plane/front) half crack is located at the Cu substrate, and the CNT was located at the opposite side of the specimen.

**Figure 2 nanomaterials-13-00575-f002:**
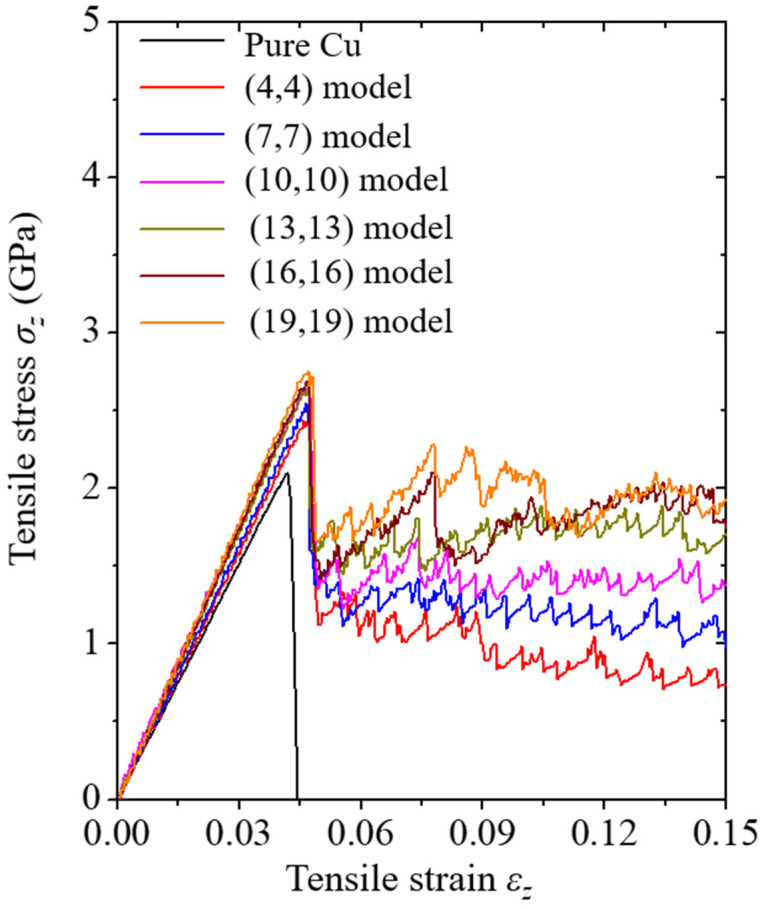
Stress–strain curves of the simulation models under tension. “Pure Cu” indicates the pure Cu specimen without CNT, and the (*n*,*n*) index indicates the chiral vector of the CNT in the simulation model.

**Figure 3 nanomaterials-13-00575-f003:**
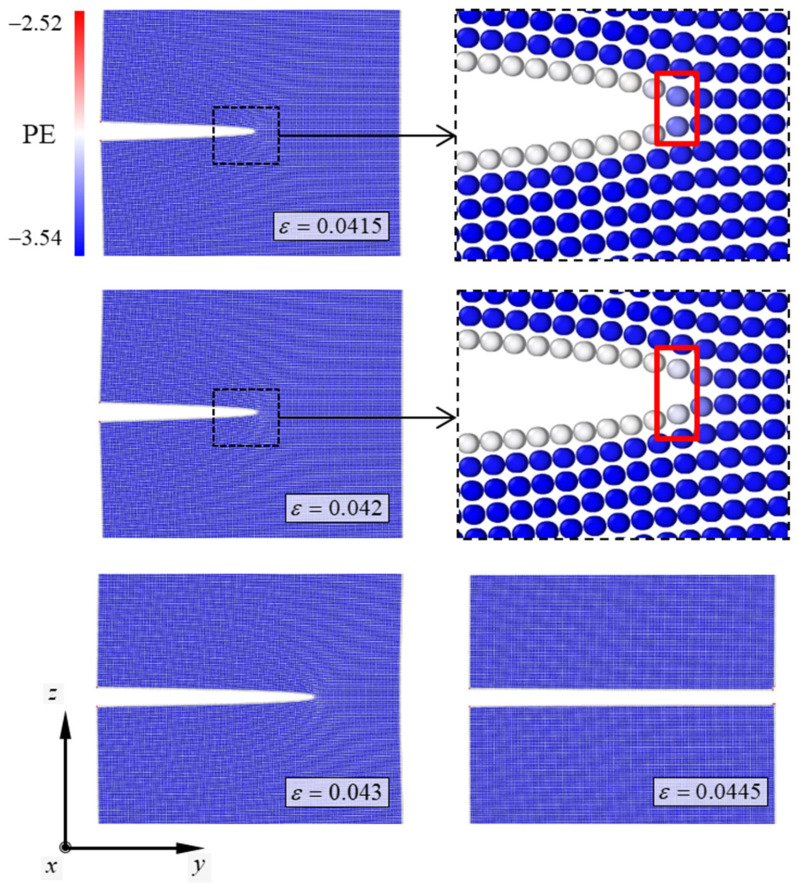
Atomic configurations depicting the fracture mechanism of the pure Cu specimen. PE denotes potential energy of the atom consisting of the simulation model. The numbers next to the color bar denote the range of the PE and the SI unit of the PE is eV. The atomic configuration in the solid line box shows the crack tip of the specimen in detail.

**Figure 4 nanomaterials-13-00575-f004:**
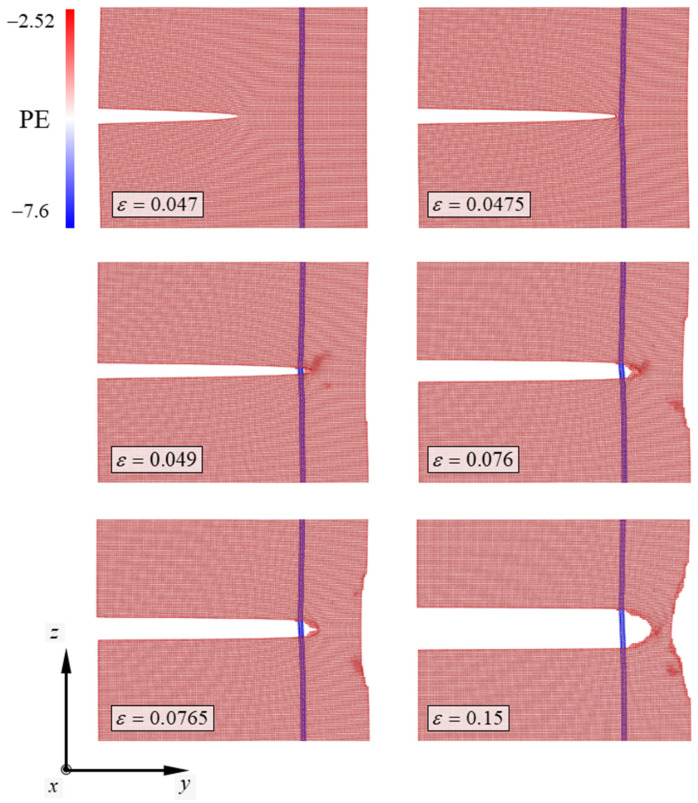
Atomic configurations depicting the deformation mechanism of the Cu/CNT composite containing the (4,4) CNT. PE denotes potential energy of the atom consisting of the simulation model. The numbers next to the color bar denote the range of the PE and SI unit of the PE is eV.

**Figure 5 nanomaterials-13-00575-f005:**
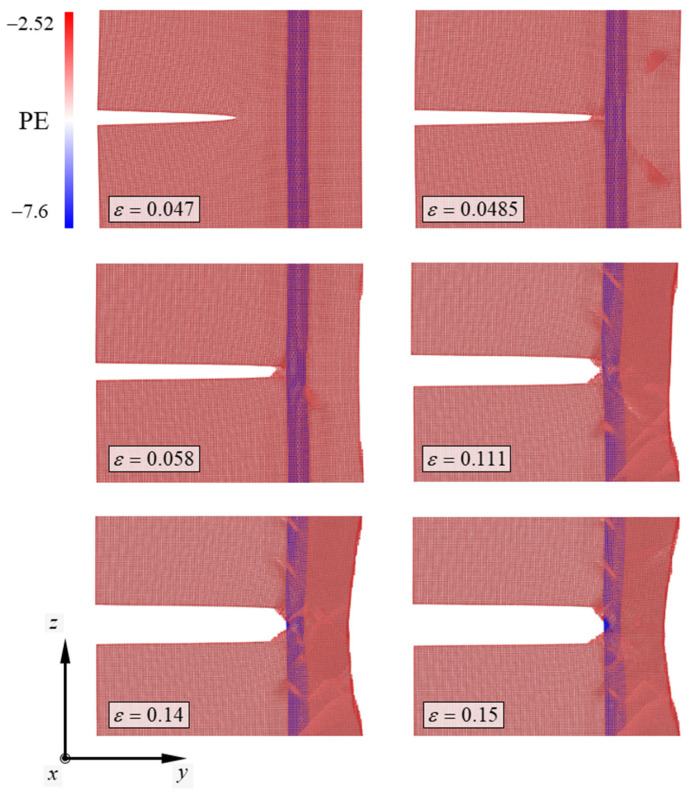
Atomic configurations depicting the deformation mechanism of the Cu/CNT composite containing the (19,19) CNT. PE denotes potential energy of the atom consisting of the simulation model. The numbers next to the color bar denote the range of the PE and SI unit of the PE is eV.

**Figure 6 nanomaterials-13-00575-f006:**
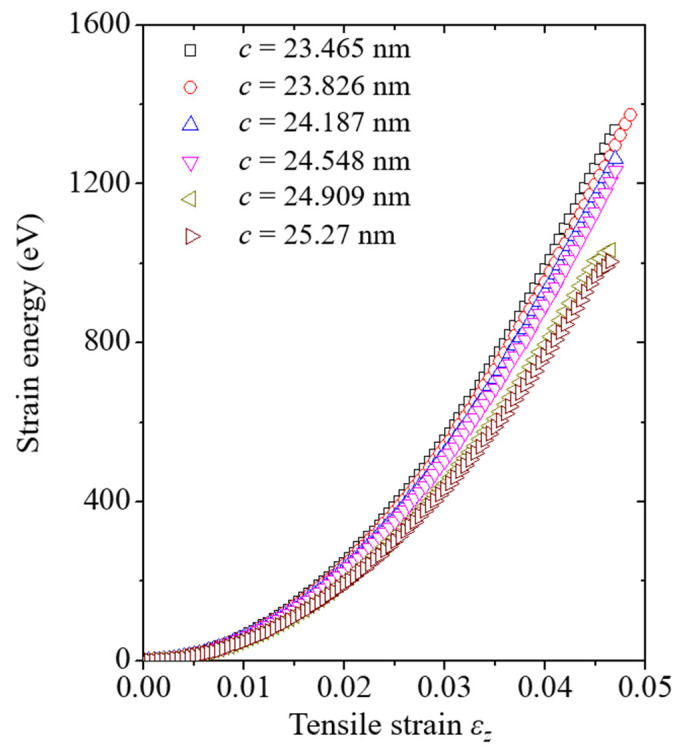
Strain energy of the (4,4) CNT model with various crack lengths corresponding to the tensile strain. *c* denotes the crack length in the model.

**Figure 7 nanomaterials-13-00575-f007:**
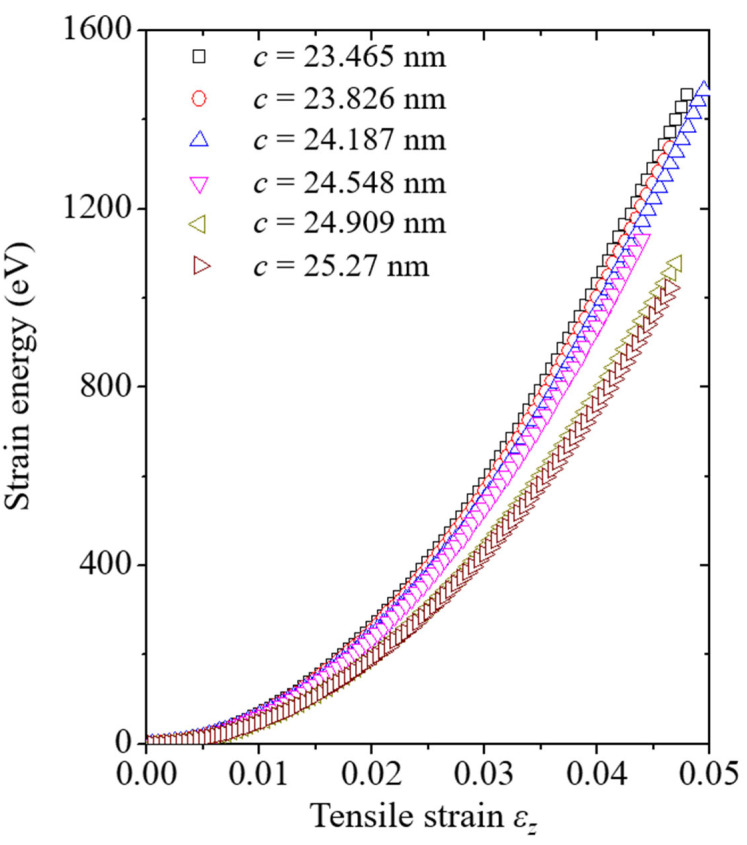
Strain energy of the (7,7) CNT model with various crack lengths corresponding to the tensile strain. *c* denotes the crack length in the model.

**Figure 8 nanomaterials-13-00575-f008:**
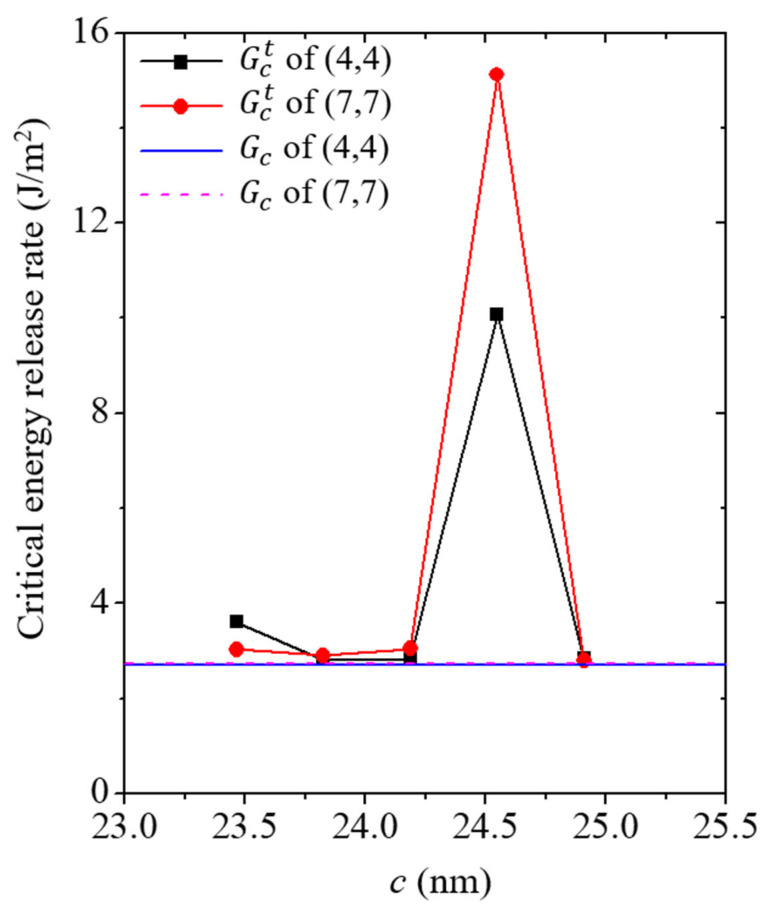
Comparison of the critical energy release rate of the simulation model between the two-specimen method and Griffith criterion. *G*_c_ denotes critical energy release rate values calculated by the Griffith criterion, which is equivalent to Equation (6). *G*_c_^t^ denotes critical energy release rate values calculated by the two-specimen method, which is equivalent to Equation (7).

**Table 1 nanomaterials-13-00575-t001:** Comparison of the stiffness and cohesive energy of the materials between simulation data and experimental values. The SI unit of the stiffness *C*_ij_ and Young’s modulus *Y* is GPa. The SI unit of the cohesive energy *E*_coh_ is eV/atom. The chiral vector of the CNT utilized in the table is (19,19).

	Cu-Cu			C-C	
	EAM [13]	Experiment [17,18]		REBO [14,15]	Experiment [19,20]
*C* _11_	170.94	170	*Y*	809.36	810 ± 410
*C* _12_	122.64	122.5			
*C* _44_	76.18	75.8			
*E* _coh_	−3.54	−3.54	*E* _coh_	−7.40	−7.41

**Table 2 nanomaterials-13-00575-t002:** Radius of the CNT corresponding to the chiral vector of the CNT utilized in this study.

Chiral Index	(4,4)	(7,7)	(10,10)	(13,13)	(16,16)	(19,19)
Radius (nm)	0.2757	0.4824	0.6892	0.8959	1.1027	1.3094

**Table 3 nanomaterials-13-00575-t003:** Young’s modulus *Y*_MS_ of the Cu/CNT composite calculated by the MS simulation and the parameters for calculating Young’s modulus *Y*_RM_ of the system according to the Equation (5). The *r* and *R* indicate the radius of the CNT and the hole in the system, respectively.

Chiral Index	(4,4)	(7,7)	(10,10)	(13,13)	(16,16)	(19,19)
*Y*_MS_ (GPa)	52.877	54.755	56.667	57.259	58.601	59.859
*r* (nm)	0.2757	0.4824	0.6892	0.8959	1.1027	1.3094
*R* (nm)	0.5	0.7	0.9	1.1	1.4	1.6
*Y*_Cu_ (GPa)	49.947	49.635	49.408	48.649	48.118	47.566
*A*_Cu_ (nm^2^)	175.148	174.394	173.389	172.132	169.776	167.891
*Y*_CNT_ (GPa)	860.82	844.7	827.86	818.4	812.8	809.36
*A*_CNT_ (nm^2^)	0.589	1.031	1.472	1.914	2.356	2.797
*Y*_RM_ (GPa)	52.665	54.306	55.962	57.114	58.583	60.051

**Table 4 nanomaterials-13-00575-t004:** Failure strength and surface energy $\gamma_{s}$ of the simulation models in this study. “Pure Cu” indicates the failure properties of the pure Cu specimen with a half nanocrack. The experiment indicates the experimental value of the Cu/CNT composite.

Chiral Index	(4,4)	(7,7)	(10,10)	(13,13)	(16,16)	(19,19)	Pure Cu	Experiment [38]
Failure Strength (GPa)	2.471	2.551	2.678	2.671	2.691	2.755	2.098	1.01 ± 0.13
$\gamma_{s}$ (J/m^2^)	1.355	1.368	1.377	1.394	1.402	1.412	1.342	

## Data Availability

The data presented in this study are available on request from the corresponding author.

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
