# Peer review of "Atomistic Investigation on the Blocking Phenomenon of Crack Propagation in Cu Substrate Reinforced by CNT"

_nanomaterials, 2023, doi:10.3390/nano13030575_

Round 1

Reviewer 1 Report

An interesting paper, but it causes multiple methodological questions

1) forcefield, in particular Cu-C. The authors discuss subtle features, like influence of the chirality on mechanical properties. At the same time, they give no description of where the parameters come from, except for ref 22, where it is stated that the model was "adopted" from some other refs (I did not make the recursive search). What properties were these LJ parameters fitted to and why this relatively simple model is adequate, remains unclear. Plus, improper referencing (papers should reference the original sources whenever possible.

BTW, I do not understand the use of ref 21. It is entirely irrelevant to the system studied

2) Not absolutely clear how the defect in the Cu crystal was made to incorporate the CNTs. I might expect that the crystal structure around the defect differs, but ho and how was if found, remains unclear

3) "Molecular statics". The problem here is that straightforward energy minimization does not allow the system to overcome even samples free energy barriers and the strain changes and the crack propagates. I would expect a certain finite-temperature relaxation after each change of the system geometry, and I am not sure that MM is fully adequate, to be honest

4) Looking at the configurations presented in Fig 4 and especially in Fig 5, I cannot help feeling that the box size should substantially affect the results. I ask authors to check this just on one single example

Overall, I think the questions should be answered and I am calling for a mandatory major revision

Regards

AV

Reviewer 2 Report

In the manuscript, the blocking phenomenon of crack propagation in Cu substrate reinforced by CNT was investigated. The present work is meaningful and interesting. However, the following questions should be considered before acceptance for publication.

  1) Would the type and size of CNT affect the performance?

  2) How to realize the uniform distribution of CNT in Cu substrate?

  3) How did the interface bonding of Cu and CNT?

  4) The properties of composite specimens should be compared to others by using a table.

Round 2

Reviewer 1 Report

I am fine with corrections